# The Neuropeptide Nocistatin Is Not a Direct Agonist of Acid-Sensing Ion Channel 1a (ASIC1a)

**DOI:** 10.3390/biom11040571

**Published:** 2021-04-13

**Authors:** Sven Kuspiel, Dominik Wiemuth, Stefan Gründer

**Affiliations:** Institute of Physiology, RWTH Aachen University, 52056 Aachen, Germany; skuspiel@ukaachen.de (S.K.); dwiemuth@ukaachen.de (D.W.)

**Keywords:** acid-sensing ion channel, neuropeptide, nocistatin

## Abstract

Acid-sensing ion channels (ASICs) are ionotropic receptors that are directly activated by protons. Although protons have been shown to act as a neurotransmitter and to activate ASICs during synaptic transmission, it remains a possibility that other ligands directly activate ASICs as well. Neuropeptides are attractive candidates for alternative agonists of ASICs, because related ionotropic receptors are directly activated by neuropeptides and because diverse neuropeptides modulate ASICs. Recently, it has been reported that the neuropeptide nocistatin directly activates ASICs, including ASIC1a. Here we show that nocistatin does not directly activate ASIC1a expressed in *Xenopus* oocytes or CHO cells. Moreover, we show that nocistatin acidifies the bath solution to an extent that can fully explain the previously reported activation by this highly acidic peptide. In summary, we conclude that nocistatin only indirectly activates ASIC1a via acidification of the bath solution.

## 1. Introduction

Acid-sensing ion channels (ASICs) are ligand-gated ion channels that are directly gated by extracellular protons [1,2]. Protons induce rapid opening of the channel; in their continued presence, ASICs completely desensitize with time constants ranging from 0.3–3 s depending on the subtype. ASICs belong to the DEG/ENaC gene family and assemble as homo- or heterotrimers; ASIC1a, ASIC1b, ASIC2 and ASIC3 are functional as homotrimers [3]. ASIC1a is the main ASIC subunit in the central nervous system (CNS) where it is widely distributed and localizes to the postsynapse [4,5]. It has been shown that ASICs mediate a fraction of the excitatory postsynaptic current in the amygdala, the nucleus accumbens, and the Calyx of Held and that protons are a genuine neurotransmitter [6,7,8]. It remains a possibility, however, that there are further agonists of ASICs independent of protons.

For example, some DEG/ENaCs from invertebrates are directly gated by neuropeptides. The FMRFamide-activated Na^+^ channel (FaNaC) from the snail *Helix* is activated by the neuropeptide FMRFamide with an EC_50_ of ~2 μM [9], Hydra Na^+^ channels (HyNaCs) are activated by Hydra-RFamides with an EC_50_ of 0.05–10 μM [10], and the myoinhibitory peptide-gated ion channel (MGIC) from the annelid *Platynereis* is activated by myoinhibitory peptides with an EC_50_ of 1–5 μM [11]. Therefore, neuropeptides are attractive candidates also for ASIC agonists. Of note, these peptide-gated ion channels desensitize only partially and slowly (with time constants of >10 s) [9,10,11].

Indeed, several neuropeptides modulate ASICs by directly binding to these ionotropic receptors. For example, dynorphin opioid neuropeptides bind to the acidic pocket of ASIC1a [12] and shift steady-state desensitization curves to higher proton concentrations [12,13]. RFamide neuropeptides bind to the central vestibule of ASIC3 [14] and delay its desensitization [14,15]. Likewise, the endogenous opioid peptides endomorphin-1 and endomorphin-2 slow desensitization of ASIC3 [16,17]. Importantly, these neuropeptides do not directly activate ASICs, but modulate the proton-gated channel. Although a screen of 109 neuropeptides with a length <20 amino acids did not identify further modulators or activators of ASICs [17], it has recently been reported that the neuropeptide nocistatin, an endogenous neuropeptide with 35 amino acids, is a direct agonist of ASIC1a, ASIC1b, ASIC2, and ASIC3 [18]. Finding a direct peptide agonist of ASICs would constitute a major breakthrough and challenge our current understanding of the role of these ion channels in the nervous system. We noted, however, that the reported affinity of nocistatin for ASIC1a is low (EC_50_ ~250 μM), which is unexpected for a physiological agonist, and that the desensitization kinetics of nocistatin-induced currents are virtually identical to the kinetics of proton-induced currents [18].

Here we revisited the direct activation of ASIC1a by nocistatin. We find that up to 500 μM nocistatin did not directly activate ASIC1a heterologously expressed in *Xenopus* oocytes or Chinese hamster ovary (CHO) cells. Nocistatin is highly acidic (10 out of the 35 amino acids are acidic; calculated pI of 3.79), however, and we found that dissolving nocistatin without adjusting the pH acidified the bath solution to an extent that is sufficient to activate ASICs with the reported potency. We therefore conclude that nocistatin indirectly activates ASIC1a via acidification of the solvent.

## 2. Materials and Methods

### 2.1. Two Electrode Voltage-Clamp in Xenopus Laevis Oocytes

ASIC1a cRNA synthesis and preparation of oocytes was described previously [11]. Briefly, oocytes were removed from anesthetized frogs; frogs were killed after final oocyte collection by decapitation. Animal care and surgery of frogs were conducted under protocols approved by the State Office for Nature, Environment, and Consumer Protection (LANUV) of North Rhine-Westphalia (NRW), Germany, and were performed in accordance with LANUV NRW guidelines. ASIC1a cRNA (0.016 ng) was injected into stage V or VI oocytes. Oocytes were kept in OR-2 medium (in mM, 82 NaCl, 2.5 KCl, 1 Na_2_HPO_4_, 4-(2-hydroxyethyl)-1-piperazineethanesulfonic acid (HEPES), 1.0 MgCl_2_, 1 CaCl_2_, and 0.5 mg/mL polyvinylpyrrolidone) at 19 °C. Two electrode voltage-clamp measurements were performed as described previously [11]. Whole cell currents were recorded 24–48 h post injection at room temperature (20–25 °C) with a TurboTec 03X amplifier (npi Electronic, Tamm, Germany). Fast solution exchange was performed by a programmable pipetting robot (screening tool; npi Electronic). Data acquisition was controlled using CellWorks 5.1.1 (npi Electronic). Data were filtered at 20 Hz and acquired at 1 kHz. Holding potential was −70 mV. Standard bath solution for two electrode voltage-clamp measurements contained (in mM) 140 NaCl, 1.8 CaCl_2_, 1 MgCl_2_, and 10 HEPES.

### 2.2. Cell Culture and Transfection

Chinese hamster ovary (CHO) cells were maintained in Ham’s Nutrient Mixture F12 (Sigma, St. Louis, MI, USA) together with 10% fetal bovine serum (FBS; Sigma, St. Louis, MI, USA) and passaged regularly. Cells were seeded and transfected after 1 day with 1 µg rat ASIC1a cDNA in pCDNA3.1 and 1 µg GFP using X-treme GENE (Sigma, St. Louis, MI, USA). Cells were patched 48 h after transfection.

### 2.3. Patch-Clamp Recordings in CHO Cells

We performed whole cell patch-clamp measurements on cultured CHO cells, using patch pipettes of 4–6 MΩ resistance filled with (in mM) 10 NaCl, 121 KCl, 2 MgCl_2_, 5 EGTA, 10 HEPES, pH 7.25. The bath solution contained (in mM) 128 NaCl, 5.4 KCl, 1 MgCl_2_, 2 CaCl_2_, 10 HEPES, pH 7.4 with NaOH. Osmolarity was adjusted with glucose to 280 mOsmol/L for the pipette solution and to 290 mOsmol/L for the extracellular solution. Data were acquired using an Axon Digidata 1440A and an Axon Axopatch 200B and recorded and analyzed using the pClamp Software (Axon Instruments, Molecular Devices, Sunnyvale, San Jose, CA, USA).

### 2.4. Peptide Synthesis and Handling

Rat nocistatin (MPRVRSVVQARDAEPEADAEPVADEADEVEQKQLQ) has been purchased from ProteoGenix (Schiltigheim, France) with a guaranteed purity of >95%. It was delivered as a lyophilized powder. For measurements with oocytes, it was dissolved in standard bath solution (pH 7.4) to obtain a concentration of 1 mM. This solution was further diluted to 0.5 mM; for one half of this solution, the pH was adjusted to pH 7.4, for the other half it was not adjusted. For measurements with CHO cells, the peptide was dissolved in bath solution to obtain a concentration of 0.25 mM; the pH was adjusted to 7.4. The solutions were freshly prepared before use.

## 3. Results and Discussion

We dissolved nocistatin in bath solution containing 10 mM HEPES with a pH of 7.4 (see Methods); this is the same buffer at the same concentration that was previously used in the study reporting activation of ASICs by nocistatin [18]. Nocistatin dissolved readily, but, as expected for a highly acidic peptide with a pI of 3.7, the pH of the solution dropped to pH values between 6.7 and 4.15 when we dissolved nocistatin to obtain a final concentration between 0.25 and 1 mM (Table 1). We titrated nocistatin-containing solutions using NaOH until we reached again neutral pH of 7.4. For comparison, we also tested a solution containing 0.5 mM nocistatin without adjustment of the pH value.

Similar to the previous study [18], we expressed rat ASIC1a in *Xenopus laevis* oocytes and performed two electrode voltage-clamp to study activation of ASIC1a by nocistatin. We stimulated ASIC1a by the application of pH 6.0 from a conditioning pH of 7.4, which elicited large transient ASIC currents. We then applied nocistatin at a concentration of 0.5 mM for 20 s in a conditioning solution, in which the pH had not been adjusted (pH 5.1), which elicited ASIC1a currents with an amplitude and kinetics similar to the currents elicited by pH 6.0 (Figure 1). Application of pH 6.0 immediately after the application of nocistatin at pH 5.1 did not elicit currents, indicating that ASIC1a was completely desensitized. Activation and concurrent desensitization of ASIC1a by a solution containing nocistatin and having pH 5.1 was expected, because of the acidic pH of this solution. In stark contrast, however, a solution containing nocistatin, in which the pH had been adjusted to pH 7.4, elicited no ASIC1a currents and application of pH 6.0 immediately after the application of nocistatin at pH 7.4 elicited ASIC currents with a full amplitude (Figure 1), indicating that nocistatin by itself can neither activate nor desensitize ASIC1a. Thus, only application of nocistatin in a solution in which the pH had not been adjusted (pH 5.1) could reproduce the action of nocistatin previously reported [18], showing that the action of nocistatin on ASIC1a is indirect via acidification of the extracellular solution. The similar current kinetics after application of acidic pH (pH 5.5) or of nocistatin was interpreted as nocistatin sharing mechanisms of activation and desensitization with protons [18]. Our results rather suggest that these mechanisms are identical because they are entirely mediated by protons.

We repeated similar experiments in CHO cells transfected with rat ASIC1a and observed the same effect as in *Xenopus* oocytes: the application of nocistatin at a concentration of 0.25 mM at neutral pH elicited no current (Figure 2) and pH 6.0 application following the nocistatin application induced canonical ASIC1a currents (Figure 2B). These data confirm that nocistatin is not an activator of ASIC1a.

We focused our study on ASIC1a, because ASIC1a is the most important ASIC subunit in the CNS and because the action of nocistatin on ASIC1a was the focus of the previous study [18]. ASIC1a is a highly sensitive proton receptor, which is fully activated by pH 6.0 [19], explaining similar current amplitudes after application of pH 6.0 or of 0.5 mM nocistatin solution with pH 5.1 (Figure 1) or after application of pH 5.5 or of 1 mM nocistatin [18]. The apparent proton affinity of ASIC1a expressed in *Xenopus* oocytes is 6.6 [19]. Strikingly, it was reported that the apparent nocistatin affinity of ASIC1a expressed in *Xenopus* oocytes is ~0.25 mM [18]. As we now show, 0.25 mM nocistatin acidifies the bath solution to pH 6.7 (Table 1), a value similar to the EC_50_ of ASIC1a for protons. This is in agreement with the idea that protons and not nocistatin activated ASICs in the previous study [18].

In addition to ASIC1a, it has been reported that 1 mM nocistatin also activated ASIC1b, ASIC2a and ASIC3 [18]. Current amplitudes elicited by 1 mM nocistatin and by pH 5.5 were similar for ASIC1b and ASIC3 [18], which is expected if this activation was mediated by protons, because ASIC1b and ASIC3 are fully activated by pH 5.5 [19,20]. The case of ASIC2a is more informative, because 1 mM nocistatin elicited eight times larger current amplitudes than moderate acidification from 7.4 to 5.5 [18]. But ASIC2a has a low proton sensitivity with an EC_50_ of ~4.0 [21,22]. Because 1 mM nocistatin acidifies the bath solution to pH 4.1 (Table 1), a larger amplitude would therefore be expected. Indeed, stronger acidification from 7.4 to 4.5 elicited currents of an amplitude approaching the one elicited by 1 mM nocistatin [18]. Thus, using ASICs as a molecular “pH meter” to estimate the pH of nocistatin-containing solutions yields values that correspond well with the values determined by us using a conventional pH meter (Table 1). In summary, although we did not study activation of ASIC1b, ASIC2a and ASIC3 by nocistatin, nocistatin-caused acidification of the bath solution could also explain the reported actions of nocistatin on these ASIC isoforms [18].

## 4. Conclusions

We were not able to reproduce a direct activation of ASC1a by nocistatin. We show, however, that all actions of nocistatin on ASICs previously reported [18] can be explained by acidification of the bath solution and, thus, by activation of ASICs through the action of protons. Our findings show that nocistatin is not a direct agonist of ASIC1a and highlight the importance of tightly controlling the final pH of solutions applied to ASICs.

## Figures and Tables

**Figure 1 biomolecules-11-00571-f001:**
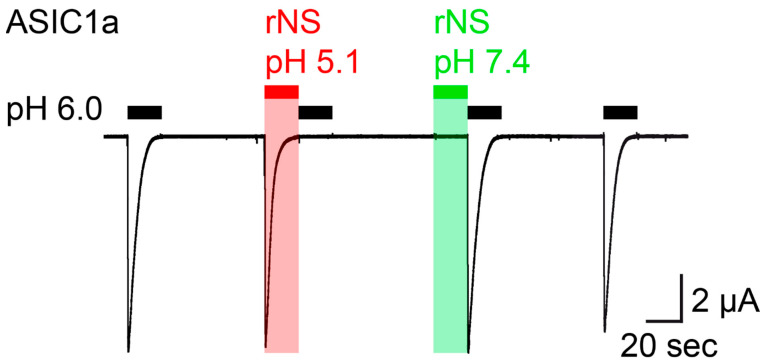
Nocistatin does not activate ASIC1a in *Xenopus* oocytes. Current trace recorded from a rASIC1a-expressing oocyte stimulated repetitively with pH 6.0 for 20 s (black bars); conditioning pH 7.4 was applied for 60 s. Rat nocistatin (rNS, 0.5 mM) in pH-unadjusted (pH 5.1, red bar) und pH-adjusted (pH 7.4, green bar) conditioning solution was applied for 20 s. This trace is representative for 5 similar measurements.

**Figure 2 biomolecules-11-00571-f002:**
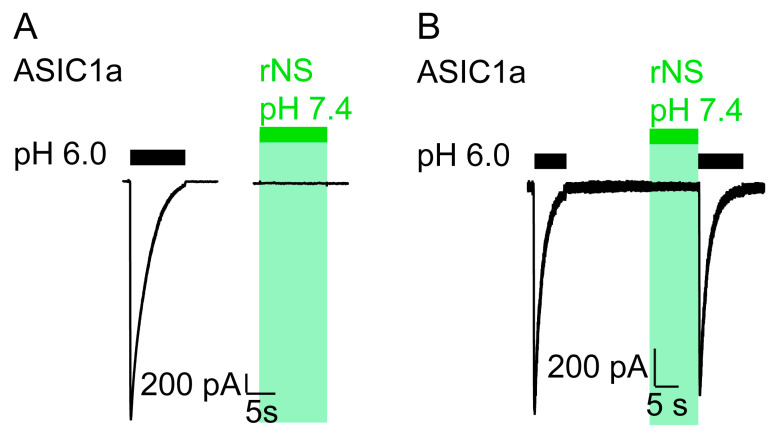
Nocistatin does not activate ASIC1a in CHO cells. (**A**) current trace of a CHO cell transfected with rASIC1a cDNA stimulated with pH 6.0 (black bar) or with rNS (0.25 mM, pH 7.4, green bar) for 10 s. Conditioning pH 7.4. This trace is representative for 6 similar measurements. (**B**) as in A, ASIC1a-expressing CHO cell was stimulated repetitively with pH 6.0 for 10 s (black bars). rNS (0.25 mM, pH 7.4, green bar) was applied for 10 s prior to pH 6.0 stimulation. This trace is representative for 3 similar measurements.

**Table 1 biomolecules-11-00571-t001:** pH values of solutions after dissolving nocistatin. Nocistatin was dissolved in a standard bath solution containing 10 mM HEPES. The pH was adjusted to 7.4 before dissolving nocistatin and the pH was then measured using a standard pH meter after dissolving different amounts of nocistatin.

Final Concentration of Nocistatin (mM)	Measured pH Value
0.25	6.7
0.5	5.1
1.0	4.2

## Data Availability

All data analyzed during this study are included in this published article.

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
