# Peer review of "The Neuropeptide Nocistatin Is Not a Direct Agonist of Acid-Sensing Ion Channel 1a (ASIC1a)"

_biomolecules, 2021, doi:10.3390/biom11040571_

Round 1

Reviewer 1 Report

This report performed a neat but rigorous assay to determine whether nocistatin gates ASIC channels. The result clearly shows that nocistatin is not a ligand of ASICs. This finding is timely given the recent report concluding nocistatin is a ligand. Instead, the current study from Grunder's group demonstrates that the previous effect can be completely explained by the acidification of the buffer by this peptide. The experiment is straightforward and the result is clean. The finding is significant to the field because it will clean a recent misleading conclusion. I therefore endorse its publication as is.

Author Response

We thank the reviewer for his positive comments.

Reviewer 2 Report

Acid-sensing ion channels (ASICs), which are activated by extracellular proton, play key roles in neuron function. Understanding its activation and regulation is essential for us to understand the function of these channels and their roles in the nervous system. It has been recently reported that the neuropeptide nocistatin can function as an agonist of ASIC channels, suggesting a new functional aspect of these channels. In the current study, Kuspiel et al. show that the activation effect of nocistatin on the ASIC1 channel is not carried by the neuropeptide itself. Instead, the effect is caused by the acidification of the bath solution when this acidic peptide is dissolved. Their experiments are straightforward, and their conclusion is clear and solid. This study has clarified the false-positive results in a previous publication and potentially avoided further confusion in the field. The manuscript is clearly written, and I do not have any big concerns.

Minor suggestions:

  1. Although nocistatin cannot directly activate the ASIC channels, it may still do so by acidification of the environment in the synaptic cleft. The authors should consider adding discussion on this possibility.
  2. If the authors have data showing nocistatin solution without pH adjustment can active ASICI in CHO cell (just like what they have shown in oocyte recording), they should add to figure 2.

Author Response

We thank the reviewer for the positive comments.

Concerning the two minor suggestions:

1. The aim of our manuscript is to transmit the clear message that nocistatin does not directly activate ASIC1a. After carefully considering to discuss the possibility that nocistatin may activate ASIC1a by acidification of the environment in the synaptic cleft, we decided not to do so because we felt this discussion would render our main message less clear. In principle, any compound acidifying the synpatic cleft could contribute to activation of ASIC1a. There is nothing special about nocistatin in this respect

2. Unfortuantely, we do not have data for CHO cells showing the effect of a nocistatin solution without pH adjustment. Due to the relatively large concentrations of nocistatin needed, we could only perform a limited amount of experiments.

Reviewer 3 Report

The authors of ‘The neuropeptide nocistatin is not a direct agonist of acid-sensing ion channel 1a (ASIC1a)’ are highlighting an important issue in biological assay development, which is the final pH of the buffer following addition of your compound of interest. They are investigating whether a recent publication reporting that nocistatin was an activator of ASIC1a can be reproduced and found that it cannot. In this case, the pI of the peptide is so low that upon dissolving the peptide in the buffer, the pH of the buffer drops to levels which activate ASICs. Ordinarily this may have gone un-noticed, but due to the nature of the ASICs , it is becoming obvious that it is the pH of the buffer solution that activates the channel and not the peptide itself. In assays where the pH of the buffer isn’t activating the channel/receptor of interest, this may have gone unnoticed, and the results could have been misinterpreted. This highlights the importance of accurate measurement of the buffers following addition of compounds prior to the biological assay. The article is well written and would be of interest to readers of Biomolecules as well as the rest of the scientific community.

No comments, publish as is.   

Author Response

(The authors gave the same response as above.)
